# Preparation of Hydrogen Peroxide Sensitive Nanofilms by a Layer-by-Layer Technique

**DOI:** 10.3390/nano8110941

**Published:** 2018-11-15

**Authors:** Kentaro Yoshida, Tetsuya Ono, Takenori Dairaku, Yoshitomo Kashiwagi, Katsuhiko Sato

**Affiliations:** 1School of Pharmaceutical Sciences, Ohu University 31-1 Misumido, Tomita-machi, Koriyama, Fukushima 963-8611, Japan; t-ono@pha.ohu-u.ac.jp (T.O.); t-dairaku@pha.ohu-u.ac.jp (T.D.); y-kashiwagi@pha.ohu-u.ac.jp (Y.K.); 2Graduate School of Pharmaceutical Sciences, Tohoku University, 6-3 Aoba, Aramaki, Aoba-ku, Sendai 980-8578, Japan; satok@m.tohoku.ac.jp

**Keywords:** hydrogen peroxide response, layer-by-layer, multilayer thin film, nanofilm, stimuli-sensitive

## Abstract

H_2_O_2_-sensitive nanofilms composed of DNA and hemin-appended poly(ethyleneimine) (H-PEI) were prepared by a layer-by-layer deposition of DNA and H-PEI through an electrostatic interaction. The (H-PEI/DNA)_5_ film was decomposed by addition of 10 mM H_2_O_2_. H_2_O_2_-induced decomposition was also confirmed in the hemin-containing (PEI/DNA)_5_ in which hemin molecules were adsorbed by a noncovalent bond to the nanofilm. On the other hand, the (PEI/DNA)_5_ film containing no hemin and the (H-PEI/PSS)_5_ film using PSS instead of DNA did not decompose even with 100 mM H_2_O_2_. The mechanism of nanofilm decomposition was thought that more reactive oxygen species (ROS) was formed by reaction of hemin and H_2_O_2_ and then the ROS caused DNA cleavage. As a result (H-PEI/DNA)_5_ and hemin-containing (PEI/DNA)_5_ films were decomposed. The decomposition rate of these nanofilms were depended on concentration of H_2_O_2_, modification ratio of hemin, pH, and ionic strength.

## 1. Introduction

Functional multilayer thin films have recently been prepared by a layer-by-layer (LbL) deposition technique through electrostatic interaction of polycation and polyanion [1,2]. Synthetic polymers [3,4], proteins [5,6,7], DNA [8,9], and polysaccharides [10,11] have been used for preparing LbL nanofilms. Such layered thin films have found application in biosensors [5,6,7], separation and purification [12,13], controlled release of drugs [14,15,16], and polyelectrolyte microcapsules [17,18,19]. In addition, stimuli-sensitive LbL films were prepared such as pH [20,21,22], sugar [23,24,25], temperature [25], ion [26], and electrical potential-sensitive LbL films [27].

In this work we report the H_2_O_2_-induced decomposition of LbL nanofilms composed of hemin-appended poly(ethyleneimine) (H-PEI) and DNA. PEI and DNA have positive and negative charge, respectively, and could form LbL multilayer film through electrostatic interaction. Hemin is iron porphyrin molecule and the active cofactor for various enzymes, such as catalase and peroxidase [28]. It is known that the iron porphyrin produces more reactive oxygen species (ROS) such as hydroxy radical (·OH) from reaction with H_2_O_2_ according to literatures [29,30]. The ROS causes non-specific DNA cleavage [31,32,33,34]. Consequently, H-PEI/DNA nanofilm was expected to decompose by addition of H_2_O_2_ as illustrated in Figure 1. On the other hand, oxidative stress of H_2_O_2_ related to various diseases caused [35,36]. Therefore, H_2_O_2_-sensitive gel and nanoparticles were used for the drug delivery system (DDS) [37,38,39,40]. It was thought that the H-PEI/DNA nanofilm is also used for DDS for this purpose by enclosing the drug inside the nanofilm. Actually, (H-PEI/DNA)_5_ film was decomposed by addition of H_2_O_2_ and the decomposition ratio depended on concentration of H_2_O_2_, modification ratio of hemin, pH, and ionic strength.

## 2. Materials and Methods

### 2.1. Materials

Hemin and DNA (calf thymus) were obtained from Sigma-Aldrich Chemical Co. (Wisconsin, WI, USA) and Funakoshi Co. (Kyoto, Japan), respectively. Poly(styrenesulfonate) sodium salt (PSS, molecular weight; 500,000) and poly(allylamine) (PAH, molecular weight; 150,000) were from Scientific Polymer Products, Inc. (Ontario, NY, USA) and Nittobo Co. (Tokyo, Japan). Poly(ethyleneimine) (PEI) was purchased from Nacalai Tesque Inc. (Tokyo, Japan). All other reagents used were of the highest grade and used without further purification.

Hemin-appended PEI (H-PEI) was synthesized as follows. PEI (100 mg) and hemin (37.9 mg) were dissolved in 0.1 M HEPES buffer (pH 7.0), then N-hydroxysuccinimide (8.02 mg) and 1-ethyl-3-(3-dimethylaminopropyl) carbodiimide hydrochloride (13.4 mg) were added in the solution at 4 °C. After 24 h, reaction mixture was purified by dialyzing to water for three days and freeze-dried. The content of hemin residues in PEI was 8.7% (molar ratio of hemin to primary amine), as determined by UV-visible absorption spectroscopy at 390 nm using a molar extinction coefficient of 42,000 M^−1^ cm^−1^ for hemin. Other modification ratio of 1.8%, 4.1%, and 6.5% H-PEI and hemin-appended PAH (H-PAH: modification ratio of 8.0%) were synthesized by similar manners and hemin was added 9.52 mg, 18.98 mg, 28.4 mg and 69.72 mg hemin to 100 mg each polymers, respectively. Chemical structures of hemin, H-PEI, H-PAH, and PSS were shown in Figure 2.

### 2.2. Apparatus

A quartz crystal microbalance (QCM, QCA 917 system, Seiko EG & G, Tokyo, Japan) was used for gravimetric analysis of the LbL films. A 9-MHz AT-cut quartz resonator coated with a thin Au layer (surface area; 0.2 cm^2^) was used as a probe, in which the adsorption of 1 ng of substance induces a −0.91 Hz change in the resonance frequency. Atomic force microscope images were recorded on SPM-9600 (SIMADZU). UV–VIS spectrum was recorded on a Shimadzu 3100PC spectrophotometer (Kyoto, Japan).

### 2.3. Preparation of Nanofilms

H-PEI/DNA nanofilms were prepared on 9 MHz Au-coated quartz resonator for QCM analysis. The quartz resonator was immersed in a H-PEI solution (0.1 mg/mL, in 0.1 M Tris-HCl buffer, pH 7.4) for 15 min to deposit the first H-PEI layer through a hydrophobic force of attraction. After being rinsed in Tris-HCl buffer for 5 min to remove any weakly adsorbed H-PEI, the quartz resonator was immersed in a DNA solution (0.1 mg/mL, in 0.1 M Tris-HCl buffer, pH 7.4) for 15 min to deposit DNA through an electrostatic interaction. The second H-PEI layer was deposited similarly on the surface of the quartz resonator. The deposition was repeated to build up nanofilms. The quartz slide was cleaned by chromic acid mixture before use. (PEI/DNA)_5_, (H-PAH/DNA)_5_ and (H-PEI/PSS)_5_ nanofilms were prepared by a same manner. Hemin-containing (PEI/DNA)_5_ and (PAH/DNA)_5_ films were prepared by immersing (PEI/DNA)_5_ and (PAH/PSS)_5_ films in 0.1 mg/mL hemin solutions into 15 min, respectively, followed by immersing 0.1 M Tris-HCl buffer for 5 min to remove any weakly adsorbed hemin.

### 2.4. Decomposition of Nanofilms

The H_2_O_2_-induced decomposition of the nanofilms was studied by monitoring the resonance frequency change (ΔF) of the 5-bilayer film-coated quartz resonator in a flow-through cell of QCM. The H_2_O_2_ solution was injected to the flow-through cell to monitor the decomposition of the nanofilm and the decomposition characteristic was evaluated by ΔF after 1 h. All experiments were carried out at room temperature (ca. 20 °C).

### 2.5. Observation of Nanofilms with AFM

For AFM observation, (H-PEI/DNA)_5_ nanofilm was prepared on a circular glass slide (diameter was 1.5 cm) which was cleaned using fuming nitric acid and thoroughly rinsed in distilled water. The glass slide was immersed in distilled water two times of 30 min to desalt and dried in a desiccator for three days. Then the film was cut with an ultrasonic cutter (USW-333, Honda Electronics Co., LTD., Aichi, Japan) and the surface and thickness of the film were observed. In addition, the (H-PEI/DNA)_5_ nanofilm was immersed in 100 mM of H_2_O_2_ solution one hour to disintegrate and observed by a similar manner. All AFM images were taken in contact mode using Olympus microcantilevers (OMCL-TR800PSA-1, Olympus Co., Tokyo, Japan) at room temperature in air.

## 3. Results and Discussion

Figure 3 shows the change in the resonance frequency (ΔF) of a QCM observed on depositing H-PEI and DNA on the quartz resonator. The ΔF values decreased on deposition of both H-PEI and DNA, showing that H-PEI and DNA could form a LbL nanofilm. It is reasonable that H-PEI and DNA form the H-PEI/DNA film, because PEI and DNA could bind through the electrostatic interaction (PEI and DNA are polycation and polyanicon, respectively). Deposition amount of H-PEI and DNA were 1.33 µg/cm^2^ and 12.8 µg/cm^2^ in (H-PEI/DNA)_5_ film. Hemin was appended to PEI at 8.7% for primary amin of PEI (PEI contains primary, secondary, and tertiary amines at 1:2:1 [41]). It was estimated that (H-PEI/DNA)_5_ film contained 6.17 µmol/cm^2^ of hemin molecules.

The H_2_O_2_-induced decomposition of a (H-PEI/DNA)_5_ film was investigated on QCM (Figure 4). ΔF has not changed in buffer solution, suggesting the (H-PEI/DNA)_5_ film was stable in this condition. By contrast, the ΔF increased when a 100 mM H_2_O_2_ solution was applied. In the case of exposing (H-PEI/DNA)_5_ film to 100 mM H_2_O_2_, ΔF was increase 1500 Hz after 1 h, and it showed that ca. 90% weight of (H-PEI/DNA)_5_ film exfoliated from the quartz slide. The increase of ΔF depended on concentration of H_2_O_2_. When (H-PEI/DNA)_5_ films were exposed to 10, 20, 50, and 100 mM H_2_O_2_ solutions, these decomposition ratios were 18.7, 22.0, 44.2, and 95.1%, respectively.

The surface morphology and thickness of (H-PEI/DNA)_5_ film was observed by AFM (Figure 5). The surface of (H-PEI/DNA)_5_ film was rough and average thickness was estimated at 27.7 nm from cross section (Figure 5a). On the other hand, the surface of (H-PEI/DNA)_5_ film was smooth and average thickness was decreased at 2.7 nm after exposure to H_2_O_2_ (Figure 5b). The thickness of (H-PEI/DNA)_5_ film was decreased by H_2_O_2_ processing. It was suggested that (H-PEI/DNA)_5_ film was decomposed by H_2_O_2_.

To investigate the mechanism of the (H-PEI/DNA)_5_ film decomposition, various polymers were used for H_2_O_2_-sensitive LBL film preparation and these films were tested (Figure 6). (H-PEH/DNA)_5_ film was decomposed at 96% by addition of 100 mM H_2_O_2_ solution after 1 h. By contrast, the decomposition rate of (PEI/DNA)_5_ which did not contain hemin molecules was 3.6%. A hemin-containing (PEI/DNA)_5_ film in which hemin was adsorbed by a noncovalent bond after preparation of film was decomposed 85.5% by H_2_O_2_. These results suggested that hemin molecules were necessary for H_2_O_2_ decomposition of nanofilms for producing of ROS from reaction with H_2_O_2_. The decomposition rate of the (H-PEI/PSS)_5_ film prepared using PSS instead of DNA and (H-PAH/DNA)_5_ film prepared using PAH instead of PEI were 2.6% and 8.4% under same conditions. It was thought that LbL nanofilms composed of PEI and DNA were convenient for H_2_O_2_ response. The ROS effectively cleaved DNA more than PSS. In addition, PEI was branched polycation and its electric charge density is lower than that of liner polycation of PAH. Low charge density of PEI was useful for degradation. Porphyrin molecules such as hemin could interact with a major groove of DNA [42,43]. It was known that PEI binding porphyrin could interact with DNA effectively than PAH binding one [43]. Therefore, (H-PEH/DNA)_5_ and hemin-containing (PEI/DNA)_5_ film were decomposed by H_2_O_2_ more effectively than other nanofilms.

The decomposition properties of (H-PEI/DNA)_5_ film were investigated under various conditions. Figure 7A shows the effect of hemin modification ratio in PEI on decomposition ratio of (H-PEI/DNA)_5_ films. The (H-PEI/DNA)_5_ film which used modification ratio of 6.5% and 4.1% H-PEI film were more sensitive than 1.8% of one, because these films could react effectively with H_2_O_2_ by lot of hemin molecules in film. However, decomposition of modification ratio of 8.7% hemin film was decreased. We thought that the interaction between hemin and DNA was excessively formed in nanofilm and disturbed the decomposition of the film. Effects of reaction solution pH and ion strength were studied (Figure 7B,C). The pH 9 solution decomposed (H-PEI/DNA)_5_ film effectively more than pH 4 and pH 7 solutions. The interaction for (H-PEI/DNA)_5_ film formation weakened, because positive charge of PEI was decreased in pH 9 solution. Thus, H-PEI and DNA interaction for composition of LbL film was decreased and LbL film was decomposed low concentration of H_2_O_2_. Similarly, high ion strength reduced electrostatic interaction of PEI and DNA for LbL film. Therefore, (H-PEI/DAN)_5_ film was decomposed by low concentration of H_2_O_2_ in 300 mM NaCl solution (Figure 7C).

The decomposition properties of hemin-containing (PEI/DNA)_5_ films were investigated similarly. Figure 8A shows the decomposition ratio of hemin-containing (PEI/DNA)_5_ films prepared by immersing in 0.1 mg/mL and 0.01 mg/mL hemin solutions to absorb on nanofilm and these films contained 2.27 nmol/cm^2^ and 1.09 nmol/cm^2^ hemin molecules, respectively. The decomposition ratio of hemin-containing (PEI/DNA)_5_ film increased with increasing hemin content in film. However, hemin-containing (PEI/DNA)_5_ film (1.0 mg/mL) and (H-PEI/DNA)_5_ (6.5%) film were 62% and 85%, respectively. It seems that introduce hemin molecules into the LbL film by covalent bond was effective for the decomposition of the LbL film. Effects of reaction solution pH and ion strength were studied (Figure 8B,C). The pH 9 solution decomposed hemin-containing (PEI/DNA)_5_ films film effectively more than pH 4 and pH 7 solutions from same reason of (H-PEI/DNA)_5_ film. However, the ionic strength did not effect on the collapse of the film. When hemin molecules adsorbed into (PEI/DNA)_5_ film, counter ions were also adsorbed. The effect of the ion strength of the reaction solution was reduced.

## 4. Conclusions

The (H-PEI/DNA)_5_ and hemin-containing (PEI/DNA)_5_ nanofilms were prepared by LbL technique and were decomposed by addition of H_2_O_2_. The decomposition depended on modification rate of hemin, concentration of H_2_O_2_, pH and ion strength of reaction solution. We have previously reported H_2_O_2_ sensitive nanofilms using the reaction of phenylboronic acid to phenol by H_2_O_2_ [44,45]. We also developed glucose and lactate stimuli responsive nanofilm combining this thin film with the enzymatic reaction of glucose oxidase [46] and lactate oxidase [47] (These enzymes produce H_2_O_2_ from each substrate). These nanofilms responded to each stimulus and decomposed within several minutes under physiological conditions, suggesting the application of insulin and anticancer drugs to DDS. However, there are some drugs that sustained release is desirable for DDS applications. The (H-PEI/DNA)_5_ and hemin-containing (PEI/DNA)_5_ nanofilms showed sustained degradation against low concentration H_2_O_2_. If the carrier containing drugs can be degraded persistently, it may be achieved as sustained drug release carrier by using this hemin base nanofilm.

## Figures and Tables

**Figure 1 nanomaterials-08-00941-f001:**
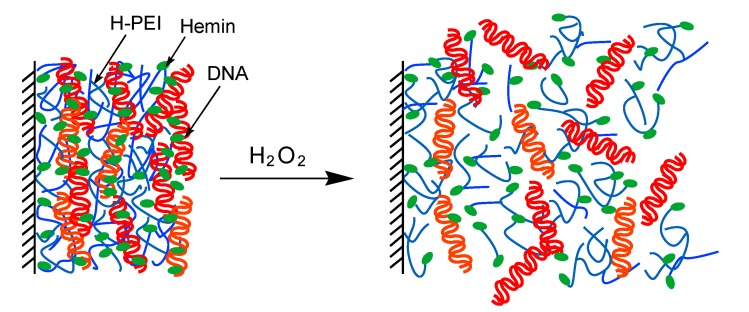
Decomposition of H-PEI/DNA nanofilm by addition of H_2_O_2_.

**Figure 2 nanomaterials-08-00941-f002:**
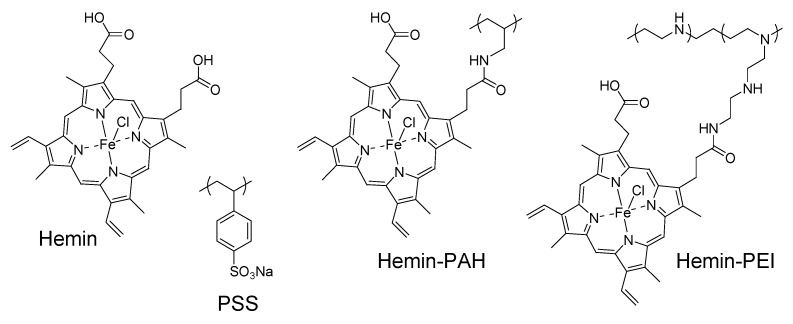
Chemical structures of hemin, H-PAH, H-PEI, and PSS.

**Figure 3 nanomaterials-08-00941-f003:**
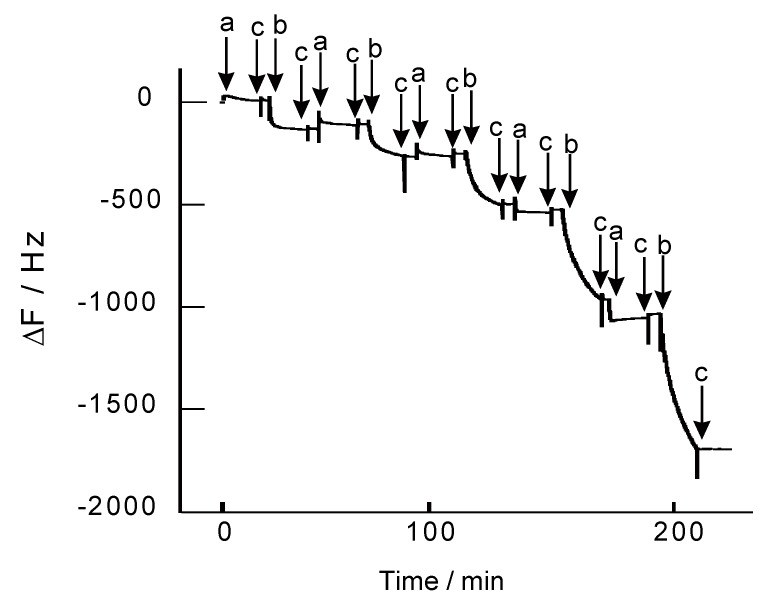
Typical QCM response for the deposition of LbL films constructed of H-PEI and DNA at pH 7.4. The quartz resonator was exposed to 0.1 mg/mL H-PEI (modification ratio of 8.7%) (**a**), 0.1 mg/mL DNA (**b**), and 0.1 M Tris-HCl buffer solution (**c**).

**Figure 4 nanomaterials-08-00941-f004:**
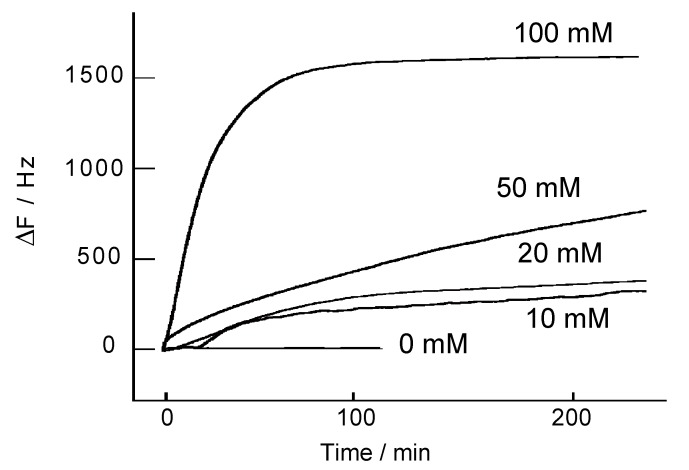
QCM frequence change of the (H-PEI/DNA)_5_ films in the presence of H_2_O_2_ in 0.1 M Tris-HCl buffer solution (pH 7.0).

**Figure 5 nanomaterials-08-00941-f005:**
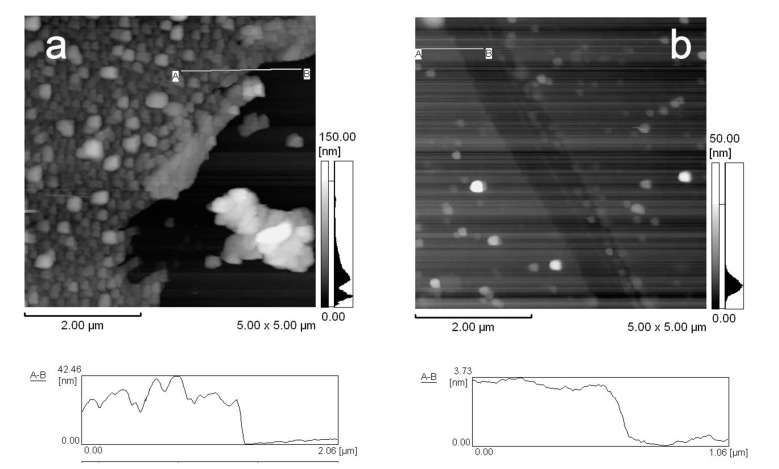
AFM images of (H-PEI/DNA)_5_ film (**a**) and after exposure to 100 mM H_2_O_2_ (**b**).

**Figure 6 nanomaterials-08-00941-f006:**
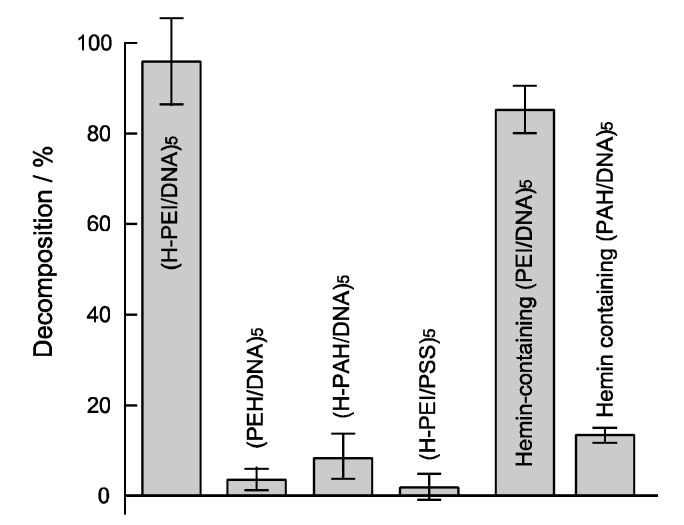
Decomposition ratio of (H-PEI/DNA)_5_, (PEI/DNA)_5_, (H-PAH/DNA)_5_, (H-PEI/PSS)_5_, hemin-containing (PEI/DNA)_5_ and hemin-containing (PAH/DNA)_5_ films after immersed in 100 mM H_2_O_2_ (0.1 M Tris-HCl buffer solution (pH 7.0) after 1 h. Modification ratio of H-PEI and H-PAH were 8.7% and 8.0%, respectively. Hemin-containing (PEI/DNA)_5_ and hemin-containing (PAH/DNA)_5_ films was prepared by immersed (PEI/DNA)_5_ and (PAH/DNA)_5_ films in 0.1 mg/mL hemin solution. Error bars represent standard deviation (*n* = 4).

**Figure 7 nanomaterials-08-00941-f007:**
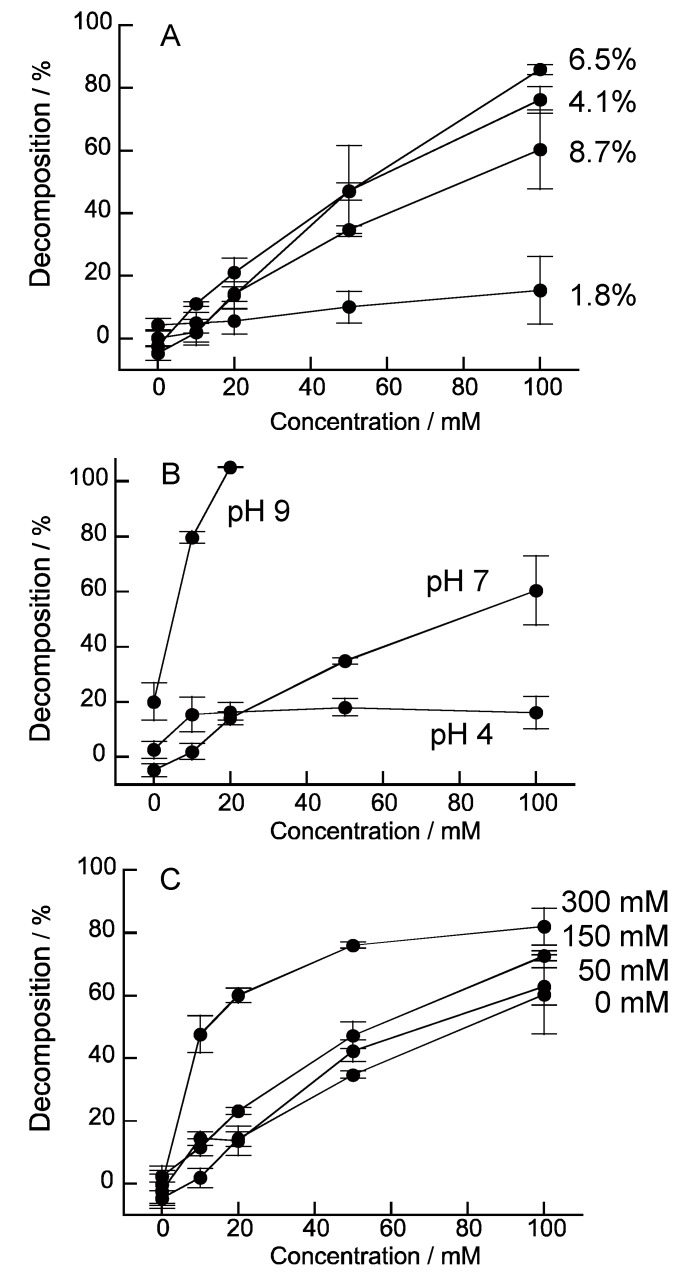
Effect of modification ratio of hemin (**A**), pH (**B**), and ion strength (**C**) on the H_2_O_2_-induced decomposition properties of (H-PEI/DNA)_5_ film. H-PEI of (**B**,**C**) were 8.7% modification ratio. Error bars represent standard deviation (*n* = 4).

**Figure 8 nanomaterials-08-00941-f008:**
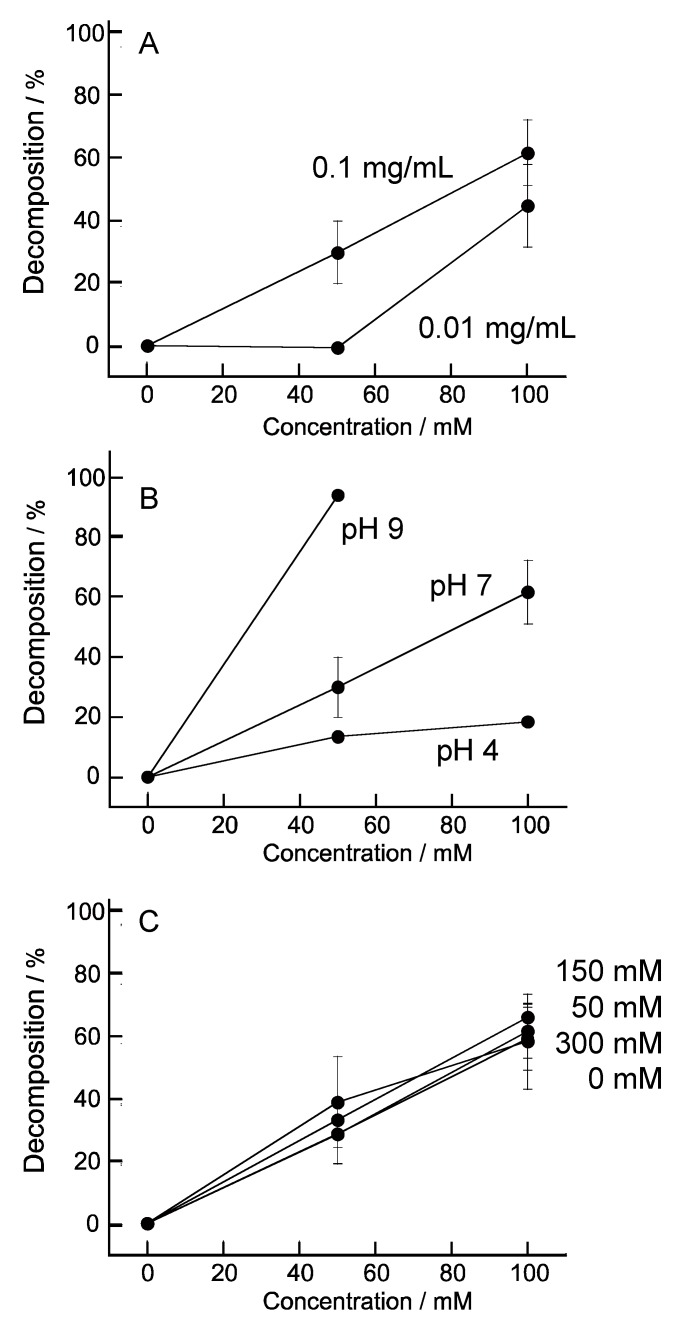
Effect of concentration of hemin solution to adsorb (**A**), pH (**B**), and ion strength (**C**) on the H_2_O_2_-induced decomposition properties of hemin-containing (PEI/DNA)_5_. Hemin-containing (PEI/DNA)_5_ of (**B**,**C**) were prepared by 0.1 mg/mL hemin solution. Error bars represent standard deviation (*n* = 4).

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
