# Peer review of "Preparation of Hydrogen Peroxide Sensitive Nanofilms by a Layer-by-Layer Technique"

_nanomaterials, 2018, doi:10.3390/nano8110941_

Round 1

Reviewer 1 Report

The manuscript is written and described aptly with a good DOE. However, going by the experience of authors it would be great to add more information about the mechanism of slow decomposition while using 8.7% hermin film (shielding effect). Also in conclusions, a note about the difference of introducing the hermin would be nice.  Apart from that, few typos and small modifications in legends e.g writing a label before its description. It is slightly confusing at times. A simple note about using solvents for QCM does not interact with the machine or the tubes can be nice. 

Author Response

Thank you for your kind review of our manuscript (nanomaterials-385643).

Reviewer 2 Report

This paper describes the preparation and studies of nanofilms capable of detection H2O2 in solution. Before publishing this paper several issues should be adressed:

Genaral

-Next time please number the lines in a continuous manner. Do not start by 1 on each page.

-You should work on your English language and style.

Page 1

Line 31: is „therefore“ a right word? The possibility to prepare LbL films is not the reason for using them as biosensors.

Line 31: “have been used to” – some word is missing. Perhaps have been used to produce? Otherwise the meaning is different: “ I am used to get up at 8 every morning”…

Line 33: “such AS”

Line 39: …H2O2 according to…

Eq. 1: something is wrong with the charges. On the left side the sum is 0 and on the right is becomes +1. Please check.

Lines 41-42: “Consequently, H-PEI/DNA nanofilm was expected to decompose by addition…”

Figure 1: there is no caption. A brief description of the Figure should be given.

Introduction in general: I miss a short explanation of the importance of your research. Why is it important to sense H2O2?

Page 2

Line 16: “as fOllows”

Page 3

Line 30-31: word order – “and the surface and thickness of the film were observed”

Line 41: appended TO

Page 4

Line 6: “HAS not changed…”

Line 7: “increased when a 100 mM H2O2 solution was applied”

Page 5

Line 9: “for producing OH…”

Page 6

Lines 6-8: I get lost in this sentence. Could you please rephrase?

Line 9: Also this sentence is unclear to me.

Line 12: “pKa of amino group was ca. pH 9”??? What does it mean?

Text on page 6: please revise it (English language, sentence structure…).

Page 7

Line 4: “hemin-containing”?

Line 8: “quantity” should be changed to “content”

Text on page 7: please also revise this passage in regard to English spelling, grammar and sentence structure.

Page 8

Line 7: WERE decomposed

Conclusion: In your conclusion you seem to talk about other results more than the ones presented. Perhaps you should just briefly mention previous work and talk about the pros of the hemin based nanofilm. Also check the spelling, grammar and sentence structure.

Author Response

Thank you for your kind suggestion of revision of our manuscript (nanomaterials-385643). We have revised the manuscript according to reviewer’s comments. All revisions made are marked in red in the revised manuscript. Our responses are as follows.

1) Next time please number the lines in a continuous manner. Do not start by 1 on each page.

[Response] The manuscript has been corrected.

2) You should work on your English language and style.

[Response] I am very sorry that my English language skill for submitting a manuscript is inadequate. We would like to encourage improvement of English language and style.

3) Page 1, Line 31: is „therefore“ a right word? The possibility to prepare LbL films is not the reason for using them as biosensors.

[Response] We agree the suggestion. The text has been corrected (p. 1, lines 31).

4) “have been used to” – some word is missing. Perhaps have been used to produce? Otherwise the meaning is different: “ I am used to get up at 8 every morning”…

[Response] The meaning is “used” in the sense of “employ” or “apply”. The text has been corrected

5) Line 33: “such AS”

[Response] The text has been corrected (p. 1, lines 33).

6) Line 39: …H2O2 according to…

[Response] The text has been corrected (p. 1, lines 39-40).

7) Eq. 1: Something is wrong with the charges. On the left side the sum is 0 and on the right is becomes +1. Please check

[Response] We agree the suggestion. According literature, hemin reacts with hydrogen peroxide to generate hydroxyl radicals and then various ROS are generated from hydroxyl radicals. Therefore, we have rewritten the description because it is difficult to describe in mathematical formulas (p. 1, lines 39-40).

8) Lines 41-42: “Consequently, H-PEI/DNA nanofilm was expected to decompose by addition…”

[Response] The text has been corrected (p. 1, lines 41-42).

9) Figure 1: There is no caption. A brief description of the Figure should be given.

[Response] We added a description of the text (Figure 1).

10) I miss a short explanation of the importance of your research. Why is it important to sense H2O2?

[Response] The text has been corrected (p. 2, lines 44-45).

11) Page 2, Line 16: “as fOllows”

[Response] The text has been corrected (p. 2, lines 58).

12) Page 3, Line 30-31: word order – “and the surface and thickness of the film were observed”

[Response] The text has been corrected (p. 3, lines 101).

13) appended TO

[Response] The text has been corrected (p. 4, lines 111).

14) Page 4, Line 6: “HAS not changed…”

[Response] The text has been corrected (p. 4, lines 119).

15) Page 4, Line 7:“increased when a 100 mM H2O2 solution was applied”

[Response] The text has been corrected (p. 4, lines 120).

16) Page 5, Line 9:“for producing OH…”

[Response] The text has been corrected (p. 5, lines 142).

17) Page 6, Lines 6-8: I get lost in this sentence. Could you please rephrase?

[Response] We improved the text as follows (p. 6, lines 161-165).

. The (H-PEI/DNA)5 film which used modification ratio of 6.5% and 4.1% H-PEI film were more sensitive than 1.8% of one, because these films could react effectively with H2O2 by lot of hemin molecules in film. However, decomposition of modification ratio of 8.7% hemin film was decreased.

18) Line 9: Also this sentence is unclear to me.

[Response] We improved the text as follows (p.6, lines 164-165).

We thought that the interaction between hemin and DNA was excessively formed in nanofilm and disturbed the decomposition of the film.

19) Line 12: “pKa of amino group was ca. pH 9”??? What does it mean?

[Response] We improved the text as follows (p.6, lines 167-168).

The interaction for (H-PEI/DNA)5 film formation weakened, because positive charge of PEI was decreased in pH 9 solution.

20) Text on page 6: please revise it (English language, sentence structure…).

[Response]  Revised in response 17)-19).

21) Page 7 Line 4: “hemin-containing”?

[Response] The description was unified to “hemin-containing”. The text has been corrected.

22) Page 7 Line 8: “quantity” should be changed to “content”

[Response] The text has been corrected (p. 7, lines 181).

23) Text on page 7: please also revise this passage in regard to English spelling, grammar and sentence structure.

[Response] We checked styles, spelling, and etc.

24) Page 8 Line 7: WERE decomposed

[Response] The text has been corrected (p. 8, lines 203).

25) conclusion: In your conclusion you seem to talk about other results more than the ones presented. Perhaps you should just briefly mention previous work and talk about the pros of the hemin based nanofilm. Also check the spelling, grammar and sentence structure.

[Response] A description was added about the future of the hemin based nanofilm and usefulness of the thin film.